# Systemic Inflammatory Indices in Second-Line Soft Tissue Sarcoma Patients: Focus on Lymphocyte/Monocyte Ratio and Trabectedin

**DOI:** 10.3390/cancers15041080

**Published:** 2023-02-08

**Authors:** Valentina Fausti, Alessandro De Vita, Silvia Vanni, Virginia Ghini, Lorena Gurrieri, Nada Riva, Roberto Casadei, Marco Maraldi, Giorgio Ercolani, Davide Cavaliere, Carlo Alberto Pacilio, Federica Pieri, Flavia Foca, Alberto Bongiovanni, Nicoletta Ranallo, Sebastiano Calpona, Giovanni Luca Frassineti, Toni Ibrahim, Laura Mercatali

**Affiliations:** 1Clinical and Experimental Oncology, Immunotherapy, Rare Cancers and Biological Resource Center, IRCCS Istituto Romagnolo per lo Studio dei Tumori (IRST) “Dino Amadori”, 47014 Meldola, Italy; 2Preclinic and Osteoncology Unit, Bioscience Laboratory, IRCCS Istituto Romagnolo per lo Studio dei Tumori (IRST) “Dino Amadori”, 47014 Meldola, Italy; 3Orthopedic Unit, “Morgagni-Pierantoni” Hospital, 47121 Forlì, Italy; 4General and Oncologic Surgery, “Morgagni-Pierantoni” Hospital, 47121 Forlì, Italy; 5Pathology Unit, “Morgagni-Pierantoni” Hospital, 47121 Forlì, Italy; 6Unit of Biostatistics and Clinical Trials, IRCCS Istituto Romagnolo per lo Studio dei Tumori (IRST) “Dino Amadori”, 47014 Meldola, Italy; 7Department of Medical Oncology, IRCCS Istituto Romagnolo per lo Studio dei Tumori (IRST) “Dino Amadori”, 47014 Meldola, Italy; 8Osteoncologia, Sarcomi dell’osso e dei tessuti molli, e Terapie Innovative, IRCCS Istituto Ortopedico Rizzoli, 40136 Bologna, Italy

**Keywords:** lymphocyte-to-monocyte ratio, prognostic marker, systemic inflammation marker, tumor microenvironment, metastatic soft tissue sarcoma

## Abstract

**Simple Summary:**

High NLR, PLR, and SII are associated with worse PFS in second-line STS patients. Trabectedin-treated patients have a better PFS when LMR is low, while patients treated with other regimens have a worse PFS when LMR is low. Patients showing a high LMR seem to have high levels of M2 intratumoral macrophages.

**Abstract:**

A second-line standard of treatment has not yet been identified in patients with soft tissue sarcomas (STS), so identifying predictive markers could be a valuable tool. Recent studies have shown that the intratumoral and inflammatory systems significantly influence tumor aggressiveness. We aimed to investigate prognostic values of pre-therapy neutrophil-to-lymphocyte ratio (NLR), platelet-to-lymphocyte ratio (PLR), lymphocyte-to-monocyte ratio (LMR), systemic inflammatory index (SII), progression-free survival (PFS), and overall survival (OS) of STS patients receiving second-line treatment. In this single-center retrospective analysis, ninety-nine patients with STS were enrolled. All patients received second-line treatment after progressing to anthracycline. PFS and OS curves were calculated using the Kaplan–Meier method of RNA sequencing, and CIBERSORT analysis was performed on six surgical specimens of liposarcoma patients. A high NLR, PLR, and SII were significantly associated with worse PFS (*p* = 0.019; *p* = 0.004; *p* = 0.006). Low LMR was significantly associated with worse OS (*p* = 0.006). Patients treated with Trabectedin showed a better PFS when the LMR was low, while patients treated with other regimens showed a worse PFS when the LMR was low (*p* = 0.0154). The intratumoral immune infiltrates analysis seems to show a correlation between intratumoral macrophages and LMR. PS ECOG. The metastatic onset and tumor burden showed prognostic significance for PFS (*p* = 0.004; *p* = 0.041; *p* = 0.0086). According to the histologies, PFS was: 5.7 mo in liposarcoma patients vs. 3.8 mo in leiomyosarcoma patients vs. 3.1 months in patients with other histologies (*p* = 0.053). Our results confirm the prognostic role of systemic inflammatory markers in patients with STS. Moreover, we demonstrated that LMR is a specific predictor of Trabectedin efficacy and could be useful in daily clinical practice. We also highlighted a possible correlation between LMR levels and the percentage of intratumoral macrophages.

## 1. Introduction

Soft tissue sarcomas are rare neoplasms of mesenchymal origin, representing about 1% of all neoplasms with an annual incidence of 3–5 cases per 100,000 inhabitants in Europe [1]. Soft tissue sarcoma (STS) currently refers to a large group of neoplasms comprising more than 80 different subgroups, depending on the tissue of origin and the type of differentiation [2]. Due to the rarity and heterogeneity of these diseases, the treatment of STS is complex and therapeutic advances have been challenging and slow [3].

While treating localized disease has progressed in recent years [4,5,6,7,8,9,10,11,12], treating metastatic disease has remained complex and challenging. Only recently have some specific histotypes received specific treatments [13]. Instead, first-line treatments have been represented by anthracyclines (with or without ifosfamide to aid tumor shrinkage) since the late 1970s [14,15,16]. There is no standard second-line therapy if the first-line therapy fails, and oncologists can choose between Trabectedin [17,18], Pazopanib [19], Eribulin [20], Gemcitabine-based regimens [21,22,23], high-dose ifosfamide [24,25,26,27], or others, depending on patient characteristics.

Considering the rarity, the heterogeneity among the various histotypes, and the lack of a therapeutic standard after the first line of treatment, it is necessary to identify prognostic factors that can help in the therapeutic decision process.

The systemic inflammatory status, represented by circulating immune cells and encoded by ratios (e.g., neutrophils/lymphocytes ratio NLR, platelets/lymphocytes ratio PLR, and lymphocytes/monocytes ratio LMR), is a prognostic index in various neoplasms [28,29,30,31]. In some studies on STS patients, the systemic inflammatory indices played a predictive role in different settings and varying histologies. Some studies showed a worse prognosis in terms of overall survival (OS), disease-specific survival (DSS), and disease-free survival (DFS) in patients with elevated NLR values. A high PLR also demonstrated a worse prognosis for OS and DFS, while patients with a low LMR showed worse OS and DFS [32,33,34,35,36,37,38,39,40,41,42].

The study’s main objective was to evaluate the prognostic role of systemic inflammatory indices in STS patients receiving second-line treatment after progression to first-line anthracycline-based treatment.

Given the peculiar activity of trabectedin on the monocyte-macrophage compartment [43,44], a secondary exploratory analysis has been performed to evaluate the predictive role of LMR in patients treated with trabectedin and to evaluate its relation to intratumoral immune infiltrate.

## 2. Material and Methods

### 2.1. Study Population

For our retrospective analysis, 99 patients with soft tissue sarcomas were included. They all received at least two lines of therapy from 2008 to 2020 at the IRCCS Istituto Romagnolo per lo Studio dei Tumori (IRST), Meldola, Italy, and all received anthracycline-based therapy in monotherapy or combination therapies as first-line patients. Patients unfit for anthracycline as first-line treatment, with concomitant chronic infection of any kind, with chronic inflammatory autoimmune disease of any kind, receiving steroids for a period longer than two weeks before the blood sample collection, or without a complete report of blood sample before second-line treatment were excluded.

Clinical biological and anatomopathological information on disease characteristics at diagnosis and information concerning the treatments performed were collected. Tumors were graded using the Fédération Nationale des Centres de Lutte Contre le Cancer (FNCLCC) system. Patients were considered oligometastatic when reporting ≤ 5 to non-symptomatic metastases. All patients underwent regular clinical and radiological follow-ups by CT, MRI, or PET FDG; radiological response to treatments was collected until death or the last follow-up according to RECIST criteria 1.1.

All follow-ups were recorded until August 2020.

### 2.2. Ethical Statement

This study was reviewed and approved by the IRST and Area Vasta Romagna Ethics Committee (approval no. 4751 of 31 July 2015); it was conducted following the Declaration of Helsinki and the Good Clinical Practice guidelines.

### 2.3. Next-Generation Sequencing Analysis

NGS analysis was performed as follows: RNA was isolated from FFPE tissue sections using the Qiagen miRNeasy FFPE kit (Qiagen, Hilden, Germany) and purified using RNeasy MinElute cleaning (Qiagen). The total RNA concentration was measured using the Qubit fluorometer (Thermo Fisher Scientific, Waltham, MA, USA), and the quality was checked using the 2100 Bioanalyzer with the RNA 6000 Nano kit (Agilent Technologies, Santa Clara, CA, USA). Library preparation was performed using QIAseq stranded total RNA library kit (Qiagen) with QIAseq Fast Select RNA Removal kit (Qiagen) for RNA depletion according to the manufacturer’s instructions and quality assessed by 2100 Bioanalyzer (Agilent). The libraries were run on the Illumina NextSeq 500 (Illumina) platform using NextSeq 500/550 High Output Kit v2.5 (75 Cycles) according to the manufacturer’s instructions. The analysis of the RNA sequencing data was processed using the Archer Analysis bioinformatics platform (v 6.2).

### 2.4. Immune Infiltrate Analysis

An evaluation of the immune infiltrate was made by analyzing the RNAseq output, using CIBERSORT. The CIBERSORT tool characterizes the cellular composition of complex tissues by their gene expression profile. A leukocyte signature matrix, called LM22, was designed and validated to assess the feasibility of leukocyte convolution from mass tumors. It contains 547 genes that distinguish 22 human hematopoietic cell phenotypes, including naïve and memory B cells, plasma cells, seven T cell types (CD8T cells, naïve CD4T cells, resting memory CD4T cells, memory activated CD4T cells, T cells follicular helper, T cells, γδ T cells), resting and activated natural killer (NK) cells, monocytes, macrophages (M0 macrophages, M1 macrophages, M2 macrophages), resting and activated dendritic cells (DC), resting and activated mast cells, eosinophils, and neutrophils. This method has been validated by flow cytometry and is used to determine the infiltration of immune cells into various malignancies (e.g., breast and colon cancer) [45].

### 2.5. Statistical Analysis

The NLR was computed as the ratio of the absolute neutrophil count to the absolute-lymphocyte count, the PLR was obtained by dividing the absolute-platelet count by the absolute lymphocyte count, and SII was calculated as platelet count × neutrophil count/lymphocyte count. Instead, the lymphocyte-to-monocyte ratio (LMR) was calculated as the ratio of lymphocyte/monocyte. Information on blood counts was collected within a week before starting treatment. Data were summarized by median and range or interquartile range or range for continuous variables and by the frequency and percentage for categorical variables. The normality assumption was checked using the Shapiro–Wilk test. Overall survival (OS) was defined as the time from the starting date of second-line therapy to the date of death from any cause. Similarly, progression-free survival (PFS) was computed from the starting date of second-line therapy to the date of disease progression. PFS and OS were reported as median values with a 95% confidence interval (95% CI). Survival curves were estimated using the Kaplan–Meier product-limit method (two-sided 95% CIs) and compared with the log-rank test. Estimated HRs with 95% CI was calculated using univariate and multivariate Cox proportional hazard models. Proportional hazard assumptions were tested using scaled Schoenfeld residuals. All *p*-values were based on two-sided testing. Statistical analysis was carried out using STATA/MP 14.0 for Windows (StataCorp LP, College Station, TX, USA).

## 3. Results

### 3.1. Patients’ Characteristics

Our single-center retrospective analysis included 99 patients with soft tissue sarcoma who received at least two lines of treatment at our institution between 2008 and 2020. Only patients who had received an anthracycline-based first line therapy, either as monotherapy or in combination, were considered for the study and were selected as homogeneously as possible.

The female population marginally exceeded the male population (56.6% vs. 43.4%) without reaching statistical significance, and the mean age at treatment was 64 years. The two most represented histotypes were leiomyosarcomas (34.4%) and liposarcomas (27.3%); a detailed list of all histological subtypes is shown in Table 1. Most of the patients included in the study (71.1%) had a histological grade III disease according to the FNCLCC score system. The limbs were the primary site for most patients (65.6%), and 33.4% of patients had metastatic disease at onset.

The main features are summarized in Table 1.

### 3.2. Progression-Free Survival and Overall Survival Based on Patient Characteristics and Treatments

There was no statistically significant difference in PFS according to gender, age of patients, histological grading at diagnosis, type of first-line chemotherapy, or type of response to first-line treatment. Due to imbalances within each group composition, PFS and OS were not evaluated at the primary site.

Patients with metastatic disease onset demonstrated a median PFS of 3.1 (95% CI 2.1–5.9) months vs. 5.2 months (95% CI 2.7–6.6) of patients with metachronous metastatic onset, *p =* 0.041.

Patients with oligometastatic disease showed better PFS than patients with disseminated metastatic disease [5.8 vs. 2.9 months, 95% (CI 2.8–9.4; 2.5–5.3), *p* = 0.0086]. Liposarcomas patients showed a median PFS of 5.7 compared to 3.8 months of patients affected by leiomyosarcoma and to 3.1 months of patients with other histologies (*p* = 0.053).

Patients with pretreatment performance status (PS) according to an ECOG scale of 0 also demonstrated better PFS than patients with baseline ECOG PS of 1 or 2 [6.5 vs. 3.2 vs. 1.4 months, 95% CI (3.3–12.6; 2.6–5.9; 0.6–2.6), *p* = 0.004].

None of the clinical features that proved significant for PFS maintained a statistical significance in Overall Survival. The PFS curves for ECOG, disease onset, tumor burden, and histology are shown in Figure 1.

Efficacy evaluation of second-line treatments in the whole population under examination did not show any statistically significant differences in PFS and OS. Response rates, median PFS, and OS by treatment type are shown in Table 2.

### 3.3. Progression-Free Survival and Overall Survival Based on Systemic Inflammatory Indices

Patients with pre-therapy NLR <2.5 had a median PFS of 4.1 mo (95% CI 2.7–7.5) compared to 2.7 mo (95% CI 2.4–5.7) of patients with pre-therapy NLR ≥2.5; *p* = 0.019.

This statistically significant difference was not conserved in OS [17.1 months (95% CI 9.7–22.7) vs. 11.4 months (95% CI 7.0–13.9) *p* = 0.125].

Regarding the PLR, patients showed a statistically significant difference in PFS but not in OS if PLR < 190; particularly, patients with pre-therapy PLR < 190 had a median PFS of 5.7 months (95% CI 2.9–7.4) compared to the median PFS of 2.5 months (95% CI 2.2–3.8) among patients with pre-therapy PLR ≥ 190, *p* = 0.004, and an OS of 13.6 months (95% CI 9.7–19.4) vs. 11.4 months (95% CI 3.8–14.3), *p* = 0.307.

Patients with pre-therapy SII < 991 had a median PFS of 4.7 months (95% CI 2.9–6.8) compared to the median PFS of 2.5 (95% CI 1.9–5.7) of patients with pre-therapy SSI ≥ 991, *p* = 0.006.

As with the other inflammatory indices, no difference was seen in OS [13.6 months (95% CI 10.0–19.1) vs. 10.7 months (95% CI 2.9–14.3) *p* = 0.417].

Patients with pre-therapy LMR <2.4 did not show any statistically significant difference in PFS compared to patients with values ≥2.4 [3.8 months (95% CI 2.5–8.0) vs. 3.3 months (95% CI 2.5–6.4) *p* = 0.228].

In OS, patients with LMR before therapy <2.4 had a median of 10.7 months (95% CI 5.3–13.0) compared to 17.1 months (95% CI 10.1–22.3) of patients with LMR ≥ 2.4 with a *p* = 0.006.

Given the different trend of LMR compared to the other indices and taking into account the peculiar activity of trabectedin on the monocyte-macrophages, we evaluated the predictive role of LMR in patients treated with trabectedin vs. all other treatments.

Trabectedin-treated patients with pre-therapy LMR <2.4 had a median PFS of 6.73 months (95% CI 2.60 to 48.40) compared with 4.2 months (95% CI 1.30 to 20.13) of patients with LMR L ≥ 2.4. Patients treated with other treatments with LMR before therapy <2.4 had a median PFS of 2.47 months (95% CI 2.07 to 17.03) vs. 3.57 months (95% CI 2.83 to 22, 80) of those who had LMR ≥ 2.4. The difference in terms of PFS was not statistically significant with a *p* = 0.0650.

Considering the trend evaluated in this analysis, the same evaluation was performed on patients treated with second- and third-line treatment, reaching a sample population of 144 patients.

Among the 144 patients receiving second- and third-line treatment, the median PFS, with a statistically significant *p* of 0.0154, was:-5.83 months (95% CI 3.070 to 48.400) in patients treated with Trabectedin and with LMR < 2.4;-3.37 months (95% CI 2.030 to 20.130) in patients treated with Trabectedin and with LMR ≥ 2.4;-2.5 months (95% CI 2.130 to 19,000) in patients treated with other treatments and with LMR < 2.4;-3.63 months (95% CI 2.930 to 22.800) in patients treated with other treatments and with LMR ≥ 2.4.

The PFS curves for NLR, PLR, SII, and MLR and OS curves for MLR are shown in Figure 2, Figure 3 and Figure 4.

To evaluate whether the results obtained from the evaluation of systemic inflammatory indices were a mirror of the intratumoral immune balance, we performed an RNAseq analysis and subsequent deconvolution via CIBERSORT on six surgical samples of patients affected by liposarcoma (two patients affected by dedifferentiated liposarcoma [LP2, LP3], three patients with myxoid liposarcoma [LP1, LP5, LP6], and one patient with pleomorphic liposarcoma LP4). Table 3 and Figure 5 show the inflammatory indices calculated on blood sampling performed the day before surgery and the percentages of each immune population present in the histological samples analyzed. As shown in Table 3, the lowest was the LMR in the blood sample before surgery, while the M2 intratumoral macrophage percentage was higher. In particular, patient LP5 had a lower LMR (0.85) and a higher percentage of intratumoral M2 (50.33%); on the contrary, patient LP6 had the highest LMR (4.94), and no M2 macrophage was detected in the CIBERSORT analysis.

## 4. Discussion

Our retrospective single-center study aimed to evaluate the predictive or prognostic role of inflammatory indices in 99 soft-tissue sarcoma patients receiving second-line treatment after anthracycline-based first-line treatment.

Among 99 patients, 32 received Trabectedin, 36 patients received Gemcitabine-based schedules, 14 patients received high-dose Ifosfamide as a continuous infusion, 9 patients received Dacarbazine, 5 patients received Pazopanib, and 3 patients received other treatments. Given the small sample size under examination, the number of different histotypes, and the variability among the various treatments, it is difficult to clearly define the most effective treatment for this setting. However, the disease control rates observed in our analysis were fairly consistent with the data shown in the literature, except for patients treated with high-dose Ifosfamide in continuous infusion, who showed higher partial response rates than those reported in the literature, i.e., 28.57% vs. 6%, while the disease progression rates are consistent with that reported in the literature, i.e., 57.14% vs. 61%, resulting in a lower percentage of stable disease (14.29% vs. 33%) [46]. In addition, our study showed median PFS and OS results similar to those reported in the literature.

The role of inflammation as a cause and consequence of cancer, and the consequent perpetuation of a “pro-tumoral” status both locally and systemically, has been significantly investigated in recent years, leading to further knowledge of the complex mechanisms that underlie the development and progression of cancer. From a linear model in which single intra-tumor mutational events were drivers of pathogenesis and disease progression, we have moved into a more complex vision that takes into account both the interactions between the tumor cell and local microenvironment (inflammatory cells, stromal cells, and stromal matrix of the tissues themselves) and the systemic interaction between tumor and the organism as a whole [47,48]. Therefore, systemic inflammatory indices are useful markers to indirectly quantify the “inflammatory burden” that the presence and progression of cancer can stimulate.

In this perspective, the Neutrophils–Lymphocytes and Platelets–Lymphocyte ratios are indirect indicators of tumor activity. The advantage of surrogate markers is the easy availability of the values deduced from a simple blood count that patients routinely perform for therapeutic purposes. The predictive and prognostic role of NLR or PLR has been extensively described in various cancers, including soft tissue sarcomas [29,30,31,31,32,48,49,50,51,52,53].

As reported in Table 4, few studies evaluated the prognostic role of inflammatory indices in sarcomas. Even if all the studies have been performed in different settings (localized, metastatic disease, or both) and in different subgroups (according to histology or kind of treatment), all of them concluded that higher levels of peripheral inflammatory markers are related to a worse prognosis.

The causes underlying this peripheral “inflamed status” are still not fully known. In the literature, the increase in platelets is known to be associated with a general inflammatory state; however, they can also mediate the growth of tumor cells, angiogenesis, and proliferation by releasing the vascular endothelial growth factor, fibroblast growth factor, and angiopoietin-1 together with other angiogenesis and tumor growth factors. Furthermore, platelets have a defining role in the tumor cell’s protection from immune elimination and in supporting the tumor metastatization process [74,75]. NLR is currently the most commonly used hematological marker of tumor-related inflammation. Neutrophils can remodel the extracellular matrix and promote angiogenesis, stimulating tumor cell migration and metastasis. Furthermore, neutrophils have a significant impact on immunity by inhibiting the cytolytic activity of lymphocytes, while tumor-infiltrating lymphocytes can limit the metastatic growth of tumor cells [76,77].

Our study confirmed the predictive role of systemic inflammatory markers; patients with high NRL PLR, or increased SII, showed a worse PFS than patients with low values of systemic inflammatory indices. Contrary to what was reported in the literature, these markers did not show a prognostic value in terms of OS. The variety of histotypes and the resulting prognoses variations could not explain the absence of statistical significance in OS.

Few studies have tried to relate peripheral immune-related markers to treatment response in STS. Shimada et al. [78] demonstrated that patients with low NLR had a worse OS when treated with Pazopanib compared with Eribulin and Trabectedin and that patients with low PLR had a better OS when treated with Eribulin.

To our knowledge, this is the first study to demonstrate the ability of LMR to predict a better PFS in patients treated with Trabectedin.

Elevated levels of tumor-associated macrophages are generally a poor prognostic factor in most cancers [79,80], including STS [81], and have shown that tumor-associated macrophages can influence tumor cell proliferation, stroma formation and dissolution, vascularization, and both pro- and antineoplastic inflammation [82]. In The Cancer Genome Atlas (TCGA) study on 206 adult soft tissue sarcomas, high levels of macrophages were revealed by gene expression signature analysis [83], particularly in dedifferentiated liposarcoma, myxofibrosarcoma, and UPS. Two further studies demonstrated an association between higher infiltration density of CD68 + or CD163 + macrophages and worse clinical outcomes [84,85].

The correlation between macrophage intratumor infiltration and circulating levels of monocytes is currently unknown. Even if the correlation between peripheral blood monocytes and TAMs is not certain, the prognostic and predictive role of the LMR score has been demonstrated in various pathologies and also in soft tissue sarcomas.

In our work, a higher value of circulating monocytes was associated with reduced overall survival, while no differences were observed in PFS in the whole population.

The absence of statistical differences could be explained by the small number of patients participating in the study and by the variety of treatments they underwent, in particular, the Trabectedin therapy (which showed direct activity on monocytes and macrophages).

To evaluate the LMR’s ability to predict Trabectedin’s efficacy, we evaluated its predictive role by dividing patients with LMR < or ≥2.4 according to the treatment performed (Trabectedin vs. Others). Contrary to what we have seen in the overall population, patients treated with Trabectedin and with high levels of pre-therapy monocytes (and therefore, with LMR <2.4) showed a better median PFS.

The role of trabectedin in modulating monocyte-macrophage activity is well known from several clinical and preclinical analyses [43]. It has been shown in vitro [44,86] that trabectedin:Has a cytotoxic effect on the monocyte-macrophage line (more than on other populations, such as the lymphocyte);Can inhibit the differentiation of monocytes into macrophages;Reduces the production of two pro-inflammatory mediators, CCL2 (responsible for recruiting monocytes to tumor sites) and IL-6 (tumor growth factor), in monocytes, macrophages, TAM, and isolated ovarian cancer cells;Acts through a mechanism of action involving modulation of the TME.

In a recently published paper, a high pretreatment neutrophil value with trabectedin was associated with a worse prognosis, consistent with what was observed for the predictive and prognostic role of NLR and PLR [87,88].

To our knowledge, our study is the first to show that in patients treated with Trabectedin, a higher pre-therapy value of monocytes correlates with a better PFS, which is contrary to what happens in other patients after second- and third-line treatment.

Taking into account the peculiar activity of trabectedin, we could suppose a higher drug activity in patients where the tumor–monocyte/macrophage interaction is more active.

To evaluate a possible correspondence between LMR < 2.4 and a high number of TAMs, we considered the preoperative LMR values in patients that received surgery for STS, of which we calculated the percentage of each tumor-infiltrating immune cells’ population. Among the six patients analyzed in our study, only one LP5 had a preoperative LMR < 2.4 [0.85]. The cybersort analyses of this sample revealed that macrophage cells with M2 polarization represented 50.33% of all tumor-infiltrating immune cells. This percentage was the highest compared to the other samples, where even with LMR values > 2.4, the trend of the M2 population was consistent with the blood ratios. Therefore, the higher the level of monocytes, the higher the levels of intratumoral M2-polarized macrophages.

This trend may be explained by the ability of the tumor to recall monocyte from the blood to be differentiated in M2 macrophages once it has arrived in the tumoral milieu.

Even if we suppose that a higher number of monocytes in the blood sample may be due to the M2 intratumoral macrophage and their chemotactic activity, further studies on the correlation between elevated values of circulating monocytes, TAMs, and response to treatment are needed to better understand the biological mechanisms underlying the clinical evidence shown in our analysis. In particular, we want to validate the predictive role of LMR on Trabectedin in a larger series to evaluate its role in specific patient subgroups (according to histology and age) to strengthen the validity of this marker in everyday therapeutic choice. Validation of prospective cohorts should be performed. We will implement a series of surgical samples to evaluate the correlation between peripheral inflammatory indices and the percentage of intratumoral single-cell fractions, to be able to validate the correlation between intratumoral M2 and circulating monocytes and to be able to evaluate the relationship between the other peripheral indices and the specific intratumor cell populations.

Collaterally, we observed, in the LP6 sample, a complete depletion of the monocyte and macrophage population and the presence of 46.14% of CD4 Naive T Cells (absent in the other samples) and 12.67% of activated NK Cells (present in smaller percentages in the other samples). This sample was the only one of all six samples tested, derived from a patient that received neoadjuvant chemotherapy with anthracyclines. The sample derived from this patient also showed PD-L1 values higher than 10% in immunohistochemistry compared to 1–5% expressed in the other samples (data not shown). These results, even if they need to be validated in a bigger series, confirm the already-known immunomodulating power of anthracyclines [89] and open the door to a possible evaluation of combination or sequential therapies with immunotherapeutic agents.

## 5. Conclusions

Our study, though limited by the number of cases and the heterogeneity of the samples, confirms the need to find more effective therapeutic strategies for the treatment of STSs. In this setting, we confirmed the important role of the systemic inflammatory state (NLR, PLR, and SII) in progression-free survival.

Furthermore, in this analysis, LMRs did not show a predictive role in the overall population, but once patients were divided according to the performed treatment, patients with high LMR and treated with Trabectedin showed a better PFS. On the contrary, the patients treated differently had worse PFS if reporting high pre-therapy monocyte values. This ratio has, therefore, been shown to be a specific predictor for treatment with Trabectedin and could be an easy-to-use index in daily clinical practice for the choice of second-line treatment.

## Figures and Tables

**Figure 1 cancers-15-01080-f001:**
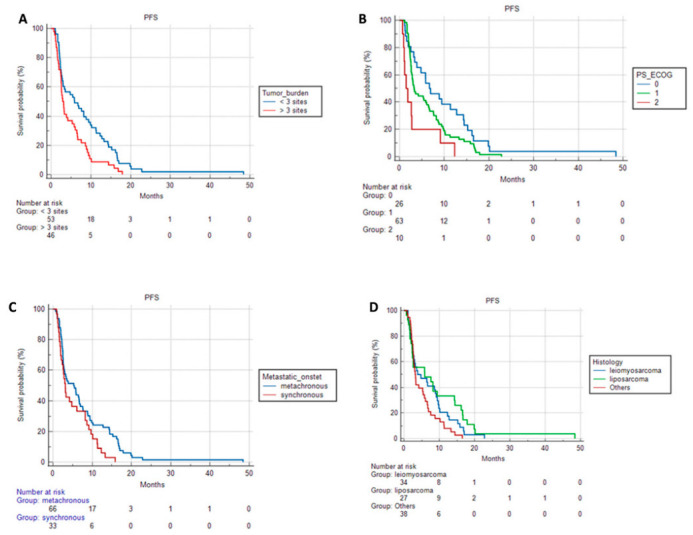
PFS according to (**A**) tumor burden; (**B**) PS ECOG; (**C**) metastatic onset; (**D**) histology.

**Figure 2 cancers-15-01080-f002:**
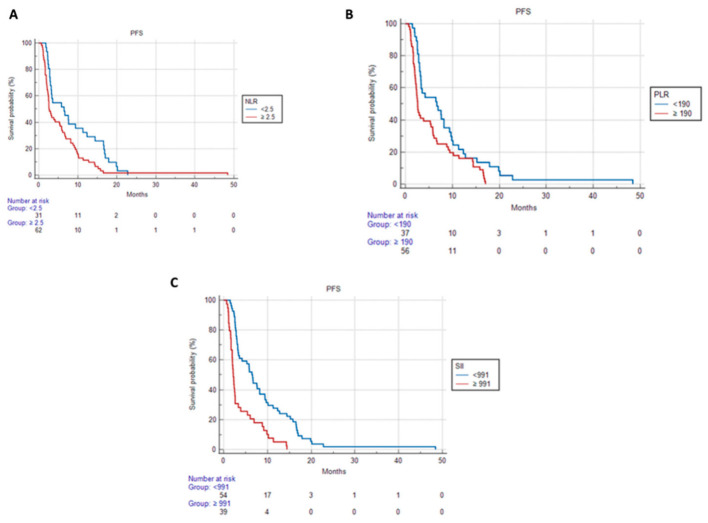
PFS according to (**A**) NLR, (**B**) PLR, and (**C**) SII.

**Figure 3 cancers-15-01080-f003:**
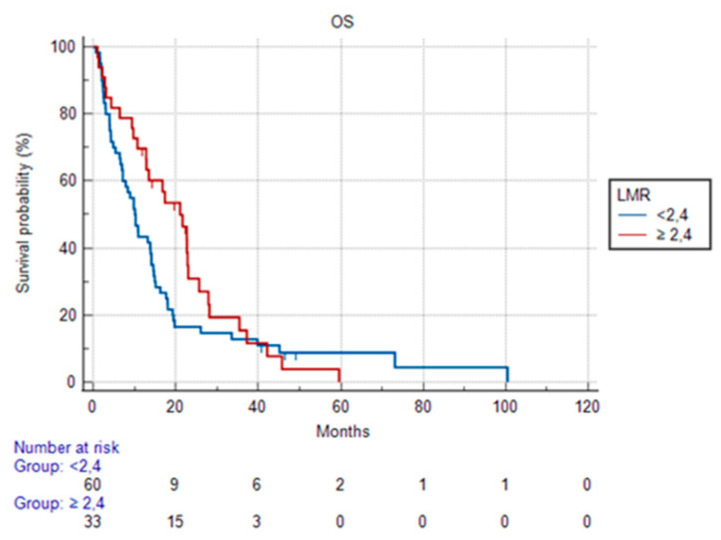
Overall survival according to LMR.

**Figure 4 cancers-15-01080-f004:**
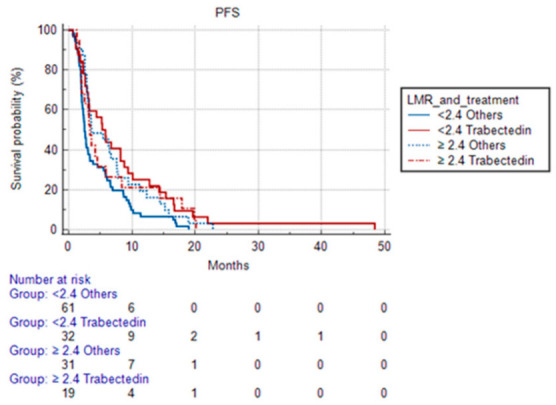
Progression-free survival according to LMR and treatment (Trabectedin vs. others).

**Figure 5 cancers-15-01080-f005:**
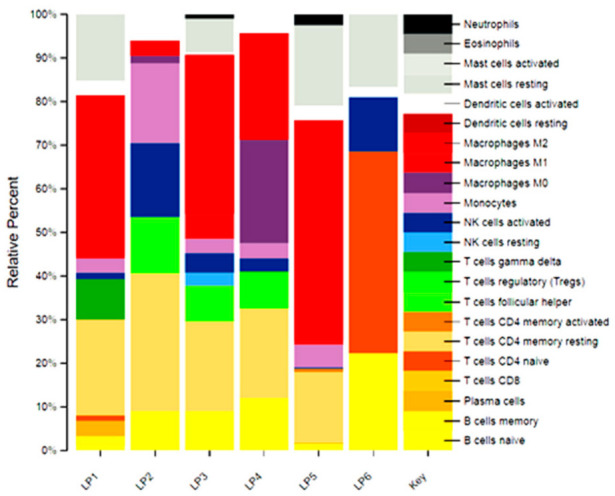
CIBESORT analysis of immune intratumoral populations and relative specific percentage.

**Table 1 cancers-15-01080-t001:** Patients’ characteristics.

Sex	
Male	43 (43.4)
Female	56 (56.6)
**Age at treatment**	
Median (range)	64 (26–83)
**Metastatic onset**	
Metachronous	66 (66.6)
Synchronous	
**Primary site**	
Retroperitoneum	8 (8)
Limbs	65(65.6)
Head and Neck	1(1.1)
Trunk	4(4.1)
Uterus	21 (21.2)
**Histotype**	
*Leiomyosarcoma*	34 (34.4)
Leiomyosarcoma nas	17
Uterine leiomyosarcoma	17
*Liposarcoma*	27 (27.3)
Well differentiated liposarcoma	7
De differentiated liposarcoma	14
Myxoid liposarcoma	2
Pleomorphic liposarcoma	4
*Others*	38 (38.3)
Angiosarcoma	7
DSRC	2
Fibrosarcoma	2
Myxofibrosarcoma	4
MPNST	1
Rhabdomyosarcoma	1
Alveolar sarcoma	1
Epithelioid sarcoma	1
Synovial sarcoma	4
High grade SES	1
SFT	4
UPS	10
**Tumor burden**	
Single metastasis	1 (1.0)
Oligometastatic	52 (53.1)
Disseminated	45 (45.9)
**Grading (FNCLCC)**	
G1	6 (6.2)
G2	22 (22.7)
G3	69 (71.1)
**PS ECOG**	
0	6 (6.2)
1	63 (63.7)
2	30 (30.1)
**First-line treatment**	
Adriamycin	39 (39.4)
Adriamycin + Dacarbazine	11 (11.1)
EI	47 (47.5)
VAI-IE	2 (2.1)

**Table 2 cancers-15-01080-t002:** Response rates, mPFS, and mOS to each second-line treatment.

II line	PD	SD	PR	TOT	mPFS (mo)	mOS (mo)
Trabectedin	11 (34.38%)	16 (50%)	5 (15.63%)	32 (100%)	6.73	13.9
Dacarbazine	7 (77.78%)	2 (22.22%)	0	9 (100%)	2.54	4.37
Gemcitabine-based	19 (52.78%)	12 (33.33%)	5 (13.89%)	36 (100%)	3.37	11.57
Ifo-HD	8 (57.14%)	2 (14.29%)	4 (28.57%)	14 (100%)	2.6	14.57
Pazopanib	3 (60%)	2 (40%)	0	5 (100%)	2.73	8.93
Others	2 (66.67%)	0	1 (33.33%)	3 (100%)	2	15.13
TOT	50 (50.51%)	34 (34.34%)	15 (15.15%)	99 (100%)		

**Table 3 cancers-15-01080-t003:** CIBESORT analysis of immune intratumoral populations and related LMR before surgery.

Mixture	LP1	LP2	LP3	LP4	LP5	LP6
NLR	2.89	2.45	3.06	2.22	6.32	3.22
PLR	142.28	125.5	147	104.22	278.76	104.22
LMR	2.94	4.87	2.96	3.23	0.85	4.94
B cells naive	0.92	9.03	8.99	11.95	1.38	22.27
B cells memory	2.35	0.00	0.00	0.00	0.00	0.00
Plasma cells	3.41	0.00	0.00	0.00	0.27	0.00
T cells CD8	0.00	0.00	0.00	0.00	0.00	0.00
T cells CD4 naive	1.36	0.00	0.00	0.00	0.00	46.14
T cells CD4 memory resting	21.96	31.58	20.61	20.52	16.43	0.00
T cells CD4 memory activated	0.00	0.00	0.00	0.00	0.56	0.00
T cells follicular helper	0.00	13.02	8.26	0.00	0.00	0.00
T cells regulatory (Tregs)	0.00	0.00	0.00	8.50	0.00	0.00
T cells gamma delta	9.33	0.00	0.00	0.00	0.00	0.00
NK cells resting	0.00	0.00	2.98	0.00	0.07	0.00
NK cells activated	1.33	17.00	4.32	3.11	0.17	12.67
Monocytes	3.33	18.01	3.38	3.49	5.36	0.00
Macrophages M0	0.00	1.83	0.00	23.58	0.00	0.00
Macrophages M1	0.69	0.00	5.78	0.00	1.13	0.00
Macrophages M2	36.75	3.61	36.40	24.55	50.33	0.00
Dendritic cells resting	0.00	0.00	0.00	0.00	0.00	0.00
Dendritic cells activated	3.27	5.92	0.59	4.29	3.41	2.25
Mast cells resting	15.29	0.00	7.62	0.00	18.31	16.68
Mast cells activated	0.00	0.00	0.00	0.00	0.00	0.00
Eosinophils	0.00	0.00	0.00	0.00	0.00	0.00
Neutrophils	0.00	0.00	1.07	0.00	2.59	0.00

**Table 4 cancers-15-01080-t004:** Published studies on inflammatory indices in sarcomas.

Ref	Author	N° of patients	Setting	Group	Index	Cutoff (≥)	Outcome	P
[54]	Garcìa-Ortega	112	Mixed	UPS	NLR	3.09	worse OS	0.04
[55]	Koseci	30	Recurrent or metastatic treated with Pazopanib	STS	NLR	3	worse PFS and OS	0.04; 0.015
[56]	Griffiths	401	Mixed	extremity STS	NLR	3	worse OS	/
[57]	Sato	141	Recurrent or metastatic	STS	NLR	3	worse OS	0.01
[58]	Yapar	172	Mixed	osteosarcoma	NLR	3.28	worse OS	<0.001
[59]	Sato	53	recurrent or metastatic treated with Eribulin	STS	NLR	3	worse	0.01
[60]	Jin	55	Localized	RMS	NLR	2.843	worse PFS and OS	0.029; 0.005
[61]	Sambri	126	Localized	MFS	NLR	3.5	worse DSS	<0.001
[62]	Netanyahu	78	Localized	RPS	NLR	2.1	worse PFS and OS	0.06, 0.3
[63]	Yamamoto	158	Advanced	STS	NLR	2.26	worse OS	/
[28]	Vinal	79	Mixed	STS	NLR	2.83	worse PFS and OS	<0.001; 0.01
[42]	Cheng	103	Mixed	SS	NLR	2.7	worse OS	0.03
[40]	Chen	42	After radical surgery	Clear Cell Sarcoma	NLR	2.73	worse OS	0.01
[64]	Mirili	26	Recurrent or metastatic treated with Pazopanib	STS	NLR	4.8	worse OS	0.02
[39]	Garcìa-Ortega	169	Presurgery	SS	NLR	3.5	worse OS	0.00
[36]	Chan	712	Localized	STS	NLR	2.5	worse PFS and OS	0.0125; 0.0112
[36]	Chan		metastatic/unresectable	STS	NLR	2.5	worse OS	0.01
[65]	Kobayashi	25	Advanced	STS	NLR	3.8	worse PFS and OS	0.001; 0.0006
[41]	Vasquez	100	Mixed	pediatric sarcomas (OS, RMS, ES)	NLR	2	worse OS	0.0237 RMS; 0.046 OS
[66]	Choi	162	Localized	STS	NLR	2.5	worse DFS	0.03
[67]	Rutkowski	385	Advanced	GIST	NLR	2.7	worse PFS and OS	<0.001; 0.001
[68]	Sobczuk	146	unresectable/metastatic GIST treated with sunitinib after failure of imatinib	GIST	NLR	2.4	worse PFS and OS	0.075; 0.002
[69]	Li	122	Mixed	ES	NLR	2.38	worse OS	0.01
[58]	Yapar	172	Mixed	OS	PLR	128	worse OS	0.01
[60]	Jin	55	Localized	RMS	PLR	162.96	worse PFS and OS	0.08; 0.05
[40]	Chen	42	After radical surgery	Clear Cell Sarcoma	PLR	103.89	worse OS	0.0147
[64]	Mirili	26	Recurrent or metastatic treated with Pazopanib	STS	PLR	195	worse OS	/
[69]	Li	122	Mixed	ES	PLR	131	worse OS	0.032
[58]	Yapar	172	Mixed	OS	LMR	4.22	Better OS	0.004
[42]	Cheng	103	Mixed	SS	LMR	4.16	Better PFS	0.025
[40]	Chen	42	After radical surgery	Clear Cell Sarcoma	LMR	4.7	Better OS	0.0445
[70]	Luo	100	Presurgery	RPLS	LMR	3	Better OS	0.002
[71]	Hou	454	After radical surgery	STS	SII	unk	worse OS	/
[72]	Ma	125	Localized	OS	SII	607.3	worse OS	/
[73]	Ouyang	86	Mixed	Pediatric OS	SII	unk	worse OS	0.05

## Data Availability

The datasets generated and/or analyzed during the current study are available from the corresponding author upon reasonable request.

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
