# Peer review of "Systemic Inflammatory Indices in Second-Line Soft Tissue Sarcoma Patients: Focus on Lymphocyte/Monocyte Ratio and Trabectedin"

_cancers, 2023, doi:10.3390/cancers15041080_

Round 1

Reviewer 1 Report (Previous Reviewer 3)

The Authors aimed to investigate the prognostic values of some inflammatory indexes for progression-free survival and overall survival of STS patients receiving a second line of treatment.

The topic is interesting. The paper was deeply ameliorated. Language should be checked by a native speaker (please provide a certificate for this control)

All acronyms should be described at first appearance. 

Methods section should include a description on how grade was calculated. How was oligometastatic/disseminated status defined?

PS EGOS...please define

I would move statistical analysis section at the end of methods.

Figures are low quality.

Is a comparison between CIBESORT and blood inflammatory cells count possible?Please discuss

Author Response

Reviewer 1

The Authors aimed to investigate the prognostic values of some inflammatory indexes for progression-free survival and overall survival of STS patients receiving a second line of treatment.

The topic is interesting. The paper was deeply ameliorated.

  1. Language should be checked by a native speaker (please provide a certificate for this control)
  2. All acronyms should be described at first appearance.
  3. Methods section should include a description on how grade was calculated.
  4. How was oligometastatic/disseminated status defined?
  5. PS EGOS...please define
  6. I would move statistical analysis section at the end of methods.
  7. Figures are low quality.
  8. Is a comparison between CIBESORT and blood inflammatory cells count possible?Please discuss

Replay: Thank you very much for your comments and suggestions: Language has been review by a native speaker; corrections have been highlighted in the text. All acronyms have been checked. Description of grading evaluation and definition of oligomestatic VS disseminated status have been added in the methods section. PS ECOG according to the international performance status scale have been specified. Statistical analysis section have been moved as requested. Comparison between CIBERSORT and blood inflammatory cell count have been discussed in the discussion.

Reviewer 2 Report (Previous Reviewer 2)

As the authors took into consideration all the reviewers' suggestions, the article might be accepted for publication in its current form.

Author Response

Reviewer 2

As the authors took into consideration all the reviewers' suggestions, the article might be accepted for publication in its current form.

Replay: Thank you very much for your comments. The reviewer has required no action.

Reviewer 3 Report (Previous Reviewer 1)

I have no further comments.

Author Response

Reviewer 3

I have no further comments.

Replay: Replay: Thank you very much for your comments. The reviewer has required no action.

Round 2

Reviewer 1 Report (Previous Reviewer 3)

The paper was significantly ameliorated according to previous comments.

This manuscript is a resubmission of an earlier submission. The following is a list of the peer review reports and author responses from that submission.

Round 1

Reviewer 1 Report

This is an excellent and well written study.

1. The authors should provide detail regarding future potential studies.

2. Do the authors plan to validate their findings.

3. Could the authors provide more detail regarding a mechanistic rationale.

4. Could this be applicable in a particular population, e.g. the elderly?

Author Response

Thank you very much for your comments and suggestions, point 1 to 4 have been added in the discussion section.

Reviewer 2 Report

This is an excellent survey regarding potential systemic inflammatory indices in second line soft tissue sarcomas patients. Nevertheless, MINOR revisions could improve article's quality.

1. In the Introduction section, the number of paragraphs should be reduced. Additional data should be commented in the Discussion section. The aim of the study should be more clear.

2. Exclusion criteria should be more analysed.

3. Figures' quality is acceptable.

4. Parallel evaluation with recent studies' evidence should be performed in the Discussion section.

5. Grammatical errors should be corrected throughout the Text. 

Author Response

Thank you very much for your comments and suggestions, the introduction section have been reduced, the aim of the study have been pointed out more clearly. Exclusion criteria have been listed in the Methods section 2.1. Parallelism with recent studies on the same topic have been implemented in the discussion section. Language check have been performed. 

Reviewer 3 Report

The Authors aimed to investigate the prognostic values of some inflammatory indexes for progression-free survival and overall survival of STS patients receiving a second line of treatment.

The topic is interesting. However, the paper is very disorganized and difficult to follow.

Introduction is too long. It is well known what STS are. Please focus on the aim of the study. 

Methods. How patients were switched to a II line ChT? How many were extremity or retroperitoneal? This should be specified as inclusion/exclusion criteria. They might have very different prognosis.

Statistical analysis section include information not strictly referred to this topic. Please correct.

Much information missing. How was grade calculated?

Any differences among retroperitoneal and extremity STS? Between infiltrative or non-infiltrative histotypes?

A careful review of previous studies reporting on inflammation markers in STS would be an added value to the discussion. Please provide a relative table.

Author Response

Thank you very much for your comments and suggestions. The introduction section have been reduced, exclusion criteria have been listed in the Methods section 2.1. The percentage of retroperitoneal or limbs or other site of origin have been listed, due to the unbalance among each arm no statistic analysis comparing different site of origin have been performed. Analysis comparing histology and grading are reported in the results section. Grading at the diagnosis have been calculated according to FNCLCC scoring system and its been specified in the text. Methods section have been checked.  An extended revision of the previous study on inflammatory indices in sarcomas have been performed, a table resuming all the previous reports have been added in the discussion section.